# The Effects of Six-Weeks Change of Direction Speed and Technique Modification Training on Cutting Performance and Movement Quality in Male Youth Soccer Players

**DOI:** 10.3390/sports7090205

**Published:** 2019-09-06

**Authors:** Thomas Dos’Santos, Alistair McBurnie, Paul Comfort, Paul A. Jones

**Affiliations:** Directorate of Sport, Exercise & Physiotherapy, University of Salford, Salford, Greater Manchester M6 6PU, UK (A.M.) (P.C.) (P.A.J.)

**Keywords:** cutting movement assessment score, movement screening, anterior cruciate ligament, change of direction deficit, injury risk profiling

## Abstract

Cutting manoeuvres are important actions associated with soccer performance and a key action associated with non-contact anterior cruciate ligament injury; thus, training interventions that can improve cutting performance and movement quality are of great interest. The aim of this study, therefore, was to determine the effects of a six-week change of dire[ction (COD) speed and technique modification training intervention on cutting performance and movement quality in male youth soccer players (U17s, n = 8) in comparison to a control group (CG) (U18s, n = 11) who continued ‘normal’ training. Cutting performance was assessed based on completion time and COD deficit, and the field-based cutting movement assessment score (CMAS) qualitative screening tool was used to assess cutting movement quality. Significant main effects for time (pre-to-post changes) (*p* ≤ 0.041, η^2^ = 0.224–0.839) and significant interaction effects of time and group were observed for cutting completion times, COD deficits, and CMASs. Improvements in completion time (*p* < 0.001, *g* = 1.63–1.90, −9% to −11% vs. −5% to 6%) and COD deficit (*p* ≤ 0.012, *g* = −1.63 to −2.43, −40–52% vs. −22% to −28%) for the COD intervention group (IG) were approximately two-times greater than the CG. Furthermore, lower CMASs (i.e., improved cutting movement quality) were only observed in the IG (*p* ≤ 0.025, *g* = −0.85 to −1.46, −23% to −34% vs. 6–19%) compared to the CG. The positive changes in CMASs were attributed to improved cutting technique and reduced incidences of high-risk deficits such as lateral trunk flexion, extended knee postures, knee valgus, hip internal rotation, and improved braking strategies. The results of this study indicate that COD speed and technique modification training, in addition to normal skills and strength training, improves cutting performance and movement quality in male youth soccer players. Practitioners working with male youth soccer players should implement COD speed and technique modification training to improve cutting performance and movement quality, which may decrease potential injury-risk.

## 1. Introduction

Soccer players, on average, perform 609 ± 193 cuts of 0° to 90° to the left or right [1] during a match, typically in response to an opponent, the ball, or to create space. Similarly, Robinson et al. [2] found 38.9 ± 13.3 and 36.3 ± 12 directional changes (45–135° movement 4 m/s or faster) per match were performed to the left and right, respectively, by soccer players. Moreover, change of direction (COD) actions (≥50°) which are then followed by a sprint are associated with critical moments, such as goal scoring and assists in soccer [3]. Consequently, given the frequency of COD actions performed in soccer, and its association with decisive moments (i.e., goal scoring), the ability to change direction rapidly can be considered an important quality to develop. 

COD speed is defined as “the ability to accelerate, reverse, or change movement direction, and accelerate again” [4], and as stated earlier, soccer players frequently perform rapid decelerations, directional changes, and sprints to create space, or to react to an opponent or ball. The determinants of COD speed are multifaceted [5] and influenced by physical (strength capacity), technical, and speed qualities [5]. Enhancements in COD speed have been demonstrated as a consequence of COD speed training interventions over 6–12 weeks (i.e., field-based sprint, deceleration, and COD drills) and are particularly effective in soccer players [6,7,8]. Moreover, a recent meta-analysis has confirmed that COD speed and sprint training interventions elicit short-term improvements in COD speed performance in soccer players [9]. Therefore, COD speed training provides practitioners with a relatively easy to perform field-based method to enhance COD performance in soccer players, requiring minimal equipment. 

While CODs are important for successful performance in soccer, directional changes are also key actions associated with non-contact anterior cruciate ligament (ACL) injury [10,11,12]. This occurrence can be attributed to the propensity to generate large multi-planar knee joint loading [13,14] during the plant foot contact, which increases ACL load [15] and can potentially rupture the ACL [16]. Moreover, greater knee joint loads are also associated with increased risk of developing patellofemoral pain (PFP) [17]; a common knee injury in soccer [18]. Despite the recent improvements in sports medicine and strength and conditioning practices in soccer, non-contact ACL injuries are not declining and are still problematic [19]. ACL injuries can be career threatening, with a plethora of negative economic, psychological, and health consequences [20,21]. Despite a high return-to-play rate in soccer following an ACL injury within a year of injury (≥90%), only two-thirds of players play at the same competitive level three years later [19]. Consequently, knee injury mitigation is of high importance for soccer players.

Non-modifiable ACL injury risk factors include hormones, anatomy, and the environment, but notably biomechanical and neuromuscular control deficits are modifiable risk factors with appropriate training [20,21]. These “high-risk” deficits during COD include the following [14,22]: wide lateral foot plant distances, greater hip abduction angles, increased internal hip and foot rotation angles, greater knee valgus, reduced knee flexion, and greater lateral trunk flexion over the plant leg. Moreover, with the exception of lateral foot plant distance and limited knee flexion [14,22], the “high-risk” COD postures offer no associated performance benefits [14], and in fact, reducing lateral trunk flexion and encouraging a trunk lean towards the direction of travel could be a faster technique [14,23,24]. COD technique modification training (i.e., coaching cues and feedback to reduce postures associated with increased knee joint loads) can address the abovementioned “high-risk” deficits and reduce potentially hazardous knee joint loading when monitored with three-dimensional (3D) motion analysis [13,25,26]. Therefore, addressing biomechanical and neuromuscular control deficits could be a viable strategy to reduce injury risk in soccer players [14,27,28]; however, no study, to date, has examined the effects of COD technique modification training on cutting movement quality in male youth soccer players.

One strategy to help reduce ACL injury risk in soccer players is evaluating movement quality to identify athletes that display “high-risk” and abnormal mechanics so that individualised training programmes can be created [20,28]. 3D motion analysis is the gold standard for evaluating movement mechanics [20] and has previously been used to monitor changes in COD mechanics [13,25,26]; however, this is a complex, time consuming, and expensive process, which is often restricted to laboratory settings, thus limiting its application in field-based settings. Recently, the cutting movement assessment score (CMAS) qualitative screening tool has been created and validated against 3D motion analysis [29,30], with strong relationships (*ρ* = 0.633–0.796, *p* < 0.001) observed between CMAS and peak knee abduction moments (KAM) (which can load the ACL) and high reliability. As practitioners will implement training interventions to reduce “high-risk” cutting mechanics, it is imperative that the effectiveness of such interventions can be monitored using a valid and reliable screening tool. The CMAS provides practitioners a valid field-based screening tool to identify athletes who generate high knee joint loads and poor movement quality; however, to the best of our knowledge, no study has monitored the effectiveness of training interventions on cutting movement quality using a field-based screening tool.

The aim of this study, therefore, was to determine the effects of a six-week COD speed and technique modification training intervention on cutting performance and movement quality in male youth soccer players, using the field-based CMAS screening tool and timing gates to assess COD quality and performance, respectively. Since the introduction of the elite performance player plan (EPPP), injury rates in adolescents have increased three-fold [31], and because youth players are striving for professional contracts, injury mitigation is paramount [31]. If athletes can reduce “high-risk” deficits (i.e., reduce the CMAS score) and improve performance (i.e., completion time and COD deficit), the COD speed and technique intervention can be considered successful. However, if athletes can reduce “high-risk” deficits while maintaining cutting performance, it can still be viewed as a positive effect. It was hypothesised that a COD speed and technique modification intervention would result in faster cutting performance and improved cutting movement quality in comparison to a CG.

## 2. Materials and Methods

### 2.1. Research Design

A non-randomized, controlled 6-week intervention study with a repeated-measures pre-to-post design was used. Youth soccer players (U17s) from an English professional soccer club were recruited for the IG, which consisted of a 6-week COD speed and technique modification training programme (Appendix A), consisting of two, 20-min sessions per week. These sessions replaced the soccer teams’ normal warm-ups for two of the sessions, which consisted of mobilisation, low-level jump-landing and sprint drills. Conversely, youth soccer players from the same club (U18s) acted as the CG and continued their normal field-based warm-ups. Pre-to-post assessments of 70° cutting COD speed performance, 10 m sprint times, and COD deficit were assessed to monitor the effectiveness of the training intervention, while cutting movement quality was assessed with the recently developed and validated CMAS screening tool [29,30].

### 2.2. Participants

A total of 26 male youth soccer players from an English professional soccer club (Under 18 s team, at the time of study, played in a regional league against youth teams of a similar standard, participated in the Football Association youth cup, and represented Manchester county; first team played in the 5th tier in the English football league at the time of the study) were recruited and participated in this study. Based on an effect size of 1.15 for pre-to-post changes (dependent *T*-Test) in COD speed performance in youth soccer players following COD speed training [7], a priori analysis, using G*Power (Version 3.1, University of Dusseldorf, Dusseldorf, Germany) [32], indicated that a minimum sample size of 8 was required to achieve a power of 0.80, and type 1 error or alpha level of 0.05. This approach was in line with recommendations for estimating sample sizes in strength and conditioning research based on effect sizes for pre-to-post designs [33]. Thirteen soccer players from the U17s squad (consisting of defenders, midfielders, and attackers) (age: 16.9 ± 0.2 years; height: 1.77 ± 0.05 m; mass: 69.2 ± 9.2 kg) were recruited for the IG. Conversely, thirteen soccer players from the U18s squad (age: 17.8 ± 0.3 years; height: 1.77 ± 0.07 m; mass: 73.3 ± 8.1 kg) acted as the CG and continued their normal field-based warm-ups (i.e., 5 min mobilisation exercises before progressing to 15 min of low-level jump-landing and sprint drills). These sample sizes were in line with those used in previous COD speed training [6,7,8] and COD technique modification studies [13,25,26]. Goalkeepers were not included in this investigation [7]. The investigation was approved by the institutional ethics review board (HSR1617-131), and all participants were informed of the benefits and risks of the investigation prior to signing an institutionally approved consent form to participate in the study.

To remain as an active participant in the study and used for further analysis, participants were not allowed to miss more than three of the 12 sessions in total (i.e., ≥75% compliance rate). Subsequently, due to match-related injuries or illness, five and two participants withdrew from the intervention and CGs, resulting in sample sizes of 8 and 11, (Figure 1) respectively. Participants in the IG completed, on average, 10.6 ± 1.2 sessions over the intervention (88.5% ± 9.9%), while the CG completed 10.8 ± 1.3 sessions (90.0% ± 10.8%) over the intervention period.

All soccer players from both groups possessed at least 5 years’ soccer training experience and had never sustained a prior knee injury such as an ACL injury prior to testing. All participants participated in the same technical and tactical soccer sessions (led by head soccer coach), five times a week (~90 min sessions on match day +2 days, +3 days, −3 days, −2 days, −1 days). Additionally, all participants performed two strength training sessions (session 1: match day +2 days, session 2; match day −3 days) a week and received the same training programmes (i.e., exercises, sets, reps, intensity). At the time of the training intervention, all players were in a strength mesocycle and played one competitive match a week. All of the procedures were carried out during the competitive season to ensure that no large physical changes were made as a result of the conditioning state [6].

### 2.3. Procedures and Field Tests

All pre-to-post testing (week 1 and week 8) was performed on the same weekday (Figure 1), at the same time of day (10:00–12:00) to control for circadian rhythm, which coincided with normal skills training time. Due to facility constraints, field testing was performed in an indoor sports hall on a hardwood court. All participants rested the day before testing and were asked to attend testing in a fed and hydrated state, similar to their normal practices before training. On arrival, all participants had their body mass (Seca Digital Scales, Model 707, Birmingham, UK) and standing height (Stadiometer; Seca, Birmingham, UK) measured to the nearest 0.1 kg and 0.1 cm, respectively. All participants then performed a standardised 10 min warm up led by the principal investigator, which consisted of dynamic stretches, low-level plyometrics, and progressive sprints drills, before completing the COD speed and 10 m sprint assessments. 

#### 2.3.1. 70° Cutting Task

The cutting task consisted of a 5 m approach, 70° cut, and 5 m exit towards the finish (Figure 2). Completion time was measured using sets of single-beam (accuracy to 1/1000th of a second) Brower photocell timing Gates (Draper, UT, USA), which were setup at the start and finish (Figure 2); time was recorded to the nearest 0.001 s. Timing gates were placed at the approximate hip height for all athletes, to ensure that only one body part breaks the beam. Participants started 0.5 m behind the first gate, to prevent any early triggering of the initial start gate, from a two-point staggered start, and were instructed to sprint as fast as possible, cut (lateral foot plant), and reaccelerate as fast as possible through the finish (Figure 2). Participants performed four practice trials at 75% of maximum perceived effort, before performing six maximum-effort cutting trials: three changing direction with a left foot plant, and three changing direction with a right foot plant, interspersed with two-minutes’ rest between trials. The testing order was counterbalanced. If the participants’ foot did not touch the cutting line (i.e., cut prematurely), slipped, turned off the incorrect foot, or did not perform a side-step cutting action, the trial was disregarded, and another trial was performed following the rest period. The mean of three trials from each limb was used for further analysis [34]. To permit qualitative screening of the cutting technique, three Panasonic Lumix FZ-200 high-speed cameras (Osaka, Japan) sampling at 100 Hz filmed the cutting trials. These cameras were positioned on tripods 3.5 m away from the cutting point at a height of 0.60 m and were placed in the sagittal and frontal plane, with a camera also placed 45° relative to the cut (Figure 2) [30].

#### 2.3.2. The 10 m Sprint

Following the cutting assessment, participants were provided with five minutes’ rest before completing the sprint assessment. Two 10 m sprint warm-up trials at 50% and 75% of maximum perceived effort were given for all participants. All participants performed three maximum-effort sprint trials, with two minutes’ rest between trials, using the same timing gates as described above placed at 0 and 10 m. Participants started 0.5 m behind the first gate, to prevent any early triggering of the initial start gate, from a two-point staggered start. The mean of the three trials was used for further analysis. 

#### 2.3.3. COD Deficit

To provide a more isolated measure of COD ability [34,35,36], COD deficit was calculated using the formula: mean 70° cutting completion time—mean 10 m sprint time [34,35]. COD deficit was calculated for left and right cutting performances.

#### 2.3.4. CMAS Screening 

Trials were screened against the 9-item CMAS screening tool [30]. If an athlete exhibited any of the characteristics/deficits, they were awarded a score, with a higher score representative of poorer technique and potentially greater peak KAMs [29,30]. All video footage was viewed in Kinovea software (0.8.15 for Windows, Bordeaux, France), which is free, and was used for qualitative screening using the CMAS. This software allowed videos to be played at various speeds and frame-by-frame. All videos were screened within two weeks pre- and post-testing. Two raters screened the videos: the principal investigator who possesses seven years’ strength and conditioning and biomechanics experience, viewed and graded each trial; the second rater was a graduate in Strength and Conditioning and possessed two years’ strength and conditioning experience. The second rater viewed and screened one trial from each participant and these scores were compared to the lead researcher to establish inter-rater reliability. Raters were allowed to independently watch the videos as many times as necessary [37,38], at whatever speeds they needed to score each test, and could also pause footage for evaluative purposes [38]. On average, qualitative screening of one cutting trial took ~3 min. Prior to qualitative screening, all raters attended a one-hour training session outlining how to grade the cutting trials using the CMAS, and to establish and uniformly agree on “low-risk” and “high-risk” movement patterns using pilot video footage. 

#### 2.3.5. Six-Week COD Speed and Technique Modification Training Intervention

A six-week COD speed and technique modification intervention, described in Appendix A, was performed by the IG twice a week (20 min per session) (session 1: match day +3 days; session 2: match day −2 days), with a minimum of 48 h between sessions. The six-week technique modification intervention focused on pre-planned low-intensity decelerations and turns (weeks 1–2), before progressing intensity via velocity and angle (weeks 3–4) [27,36,39], and introducing a stimulus with increased intensity (weeks 3–6). The COD programme was in accordance with COD speed recommendations from the National Strength and Conditioning Association [40], Nimphius [5], and recent braking-strategy recommendations [41], and the duration, distances, and number of CODs were similar to previously successful 6-week COD speed [6,7,8] and COD technique modification studies [13,25]. The sessions were led by the principle investigator, who is a certified strength and conditioning specialist with extensive experience in coaching COD speed and agility drills. All COD speed and technique modifications sessions took place at the soccer team’s training facility, with the first session of the week performed on a synthetic astro-turf and the second session of the week performed on a synthetic 3G rubber crumb field-turf. Players were given individual feedback regarding their technique, and external verbal coaching cues such as “slam on the brakes” (to promote early braking), “push/punch the ground away” (to enhance medio-lateral force propulsion and encourage active knee flexion), and “face towards the direction of travel” (to reduce lateral trunk flexion) were used to promote safer mechanics [42], promote faster performance [43], and facilitate better retention [42].

### 2.4. Statistical Analyses

All statistical analyses were performed using SPSS v 25 (SPSS Inc., Chicago, IL, USA) and Microsoft Excel (version 2016, Microsoft Corp., Redmond, WA, USA). The primary outcome variables of this study were cutting completion time, COD deficit, 10 m sprint times, and CMASs.

Within-session reliability for the primary outcome variables was assessed for each group and session using Intraclass correlation coefficients (ICC) (two-way mixed effects, average measures, absolute agreement), coefficient of variation (CV %), and standard error of measurement (SEM). The CV % was calculated as SD/mean ×100 for each participant and averaged across participants. The SEM was calculated using the formula: SD (pooled) × $\sqrt{\left(1-\mathrm{ICC}\right)}$ [44], whereas the smallest detectable difference (SDD) was calculated from the formula: (1.96 × $(\sqrt{\left(2\right)}$) × *SEM* [45]. ICCs were interpreted based on the following scale presented by Koo and Li [46]: poor (<0.50), moderate (0.50–0.75), good (0.75–0.90), and excellent (>0.90). Minimum acceptable reliability was determined with an ICC >0.7 and CV < 15% [47]. 

To determine intra-rater reliability, 26 trials (1 from trial from each participant) were randomly selected by the lead researcher, and each trial was viewed and graded on two separate occasions, separated by 7 days, in line with previous research [38]. Similarly, for inter-rater reliability, 19 trials were screened by the other researcher and these scores were compared to the lead researcher. ICCs and SEMs for total score were determined. For each item within the CMAS, percentage agreements (agreements/agreements + disagreements × 100) and Kappa coefficients were calculated [29]. Kappa coefficients were calculated using the formula: *k* = Pr(a) − Pr(e)/1 − Pr(e), where Pr(a) = relative observed agreement between raters; Pr(e) = hypothetic probability of chance agreement, which describes the proportion of agreement between the two methods after any agreement by chance has been removed [48]. The kappa coefficient was interpreted based on the following scale of Landis and Koch [49]: slight (0.01–0.20), fair (0.21–0.40), moderate (0.41–0.60), good (0.61–0.80), and excellent (0.81–1.00). Percentage agreements were interpreted in line with previous research [37] and the scale was as follows: excellent (>80%), moderate (51–79%), and poor (≤50%) [37].

Normality was inspected for all variables using a Shapiro–Wilks test. A two-way mixed ANOVA (group; time) with group as a between-subjects factor measured at 2 levels (IG and CG), and time (pre- and post-training measures) the within-subject factor. This was used to identify any significant main (time) or interaction (group × time) effects for primary outcome variables between IG and CGs. A Bonferroni-corrected pairwise comparison design was used to further analyse the effect of the group when a significant interaction effect was observed for time and group. Partial eta squared effect sizes were calculated for all ANOVAs with the values of 0.010–0.059, 0.060–0.149, and ≥0.150 considered as small, medium, and large, respectively, according to Cohen [50]. Further, pre-to-post changes in primary outcomes for each group were assessed using paired-sample t-tests for parametric data and Wilcoxon-sign ranked tests for non-parametric data. Magnitudes of differences were assessed using Hedges’ *g* effect sizes, mean change, and percentage change ((post-pre)/pre × 100) with 95% confidence intervals (CI). Hedges’ *g* effect sizes were calculated as described previously [51] and interpreted as trivial (≤0.19), small (0.20–0.59), moderate (0.60–1.19), large (1.20–1.99), very large (2.0–3.99), and extremely large (≥4.00) [52]. Comparisons in pre- and post-intervention primary outcomes and change in primary outcomes between IG and CGs were also assessed using independent sample t-tests or Mann–Whitney U tests, with effect sizes outlined above. Statistical significance was defined *p* ≤ 0.05 for all tests.

## 3. Results

### 3.1. Reliability

Within-session reliability for the IG and CG pre- and post-intervention primary outcomes are presented in Table 1 containing ICCs, CV %, SEM, and SDD. All variables for the IG displayed good to excellent ICCs pre- and post-intervention (Table 1). Cutting completion times and 10 m sprint times displayed low levels of variance pre- and post-intervention. COD deficits and CMASs displayed high levels of variance pre- and post-intervention (Table 1). All variables for the CG displayed moderate to excellent ICCs pre and post-intervention, excluding left cut completion time and COD deficit pre-intervention, which displayed lower ICCs (Table 1). Cutting completion times and 10 m sprint times displayed low levels of variance pre- and post-intervention. COD deficits and CMASs displayed high levels of variance pre- and post-intervention (Table 1).

Intra- and inter-rater percentage agreements and Kappa coefficients for intra-and inter-rater reliability are presented in Table 2.Excellent intra- (ICC = 0.972, SEM = 0.32, SDD = 0.88) and inter-rater reliability (ICC = 0.917, SEM = 0.38, SDD = 1.05) was observed for CMAS total score. Excellent intra- (≥92.3%, *k* ≥ 0.866) and inter-rater (≥89.5% *k* ≥ 0.872) percentage-agreements and kappa coefficients were demonstrated for all CMAS variables (Table 2), with the exception of penultimate foot contact (PFC) braking (*k* = 0.755) and initial knee valgus position (*k* = 0.789) which demonstrated good kappa coefficients. 

Pre-to-post changes in primary outcomes for the IG and CG are presented in Table 3 and Table 4, containing descriptives, *p* values, effect sizes, percentage differences, and mean differences. No significant differences were observed in primary outcome variables between groups pre-intervention (*p* > 0.05); however, effect sizes indicated that the IG displayed faster right cut (*p* = 0.468, *g* = −0.33), left cut (*p* = 0.097, *g* = −0.78), right COD deficit (*p* = 0.432, *g* = −0.36), left COD deficit (*p* = 0.062, g = −0.77) and greater right (*p* = 0.171, *g* = 0.71) and left CMASs (*p* = 0.456, *g* = 1.09) compared to the CG. Trivial and non-significant (*p* = 0.844, *g* = −0.09) differences were observed in 10 m sprint times between groups.

#### 3.1.1. Right Cut

Large and significant main effects for time were found for right cut completion time (*p* < 0.001, η^2^ = 0.829, power = 1.000). In addition, a large and significant interaction effect of time and group for right cut completion time was also observed (*p* = 0.010, η^2^ = 0.330, power = 0.779), with the IG showing significantly faster post-intervention completion times (*p* < 0.001, *g* = −1.94) compared to the CG. Moreover, large and significant improvements in right cut completion times were observed for the IG (*p* < 0.001, *g* = −1.90, −11.7%, −0.279 s) and CG (*p* < 0.001, *g* = −1.21, −5.9%, −0.144 s) (Table 3 and Table 4, Figure 3) post-intervention, which were greater than the SEM and SDD. Mean (*p* = 0.010, g = −1.29) and percentage (*p* = 0.005, *g* = −1.41) improvements were significantly greater for the IG compared to CG, with large effect sizes (Figure 3). 

#### 3.1.2. Left Cut

Large and significant main effects for time were found for left cut completion time (*p* < 0.001, η^2^ = 0.763, power = 1.000). In addition, a large, yet non-significant interaction effect of time and group for left cut completion time was also observed (*p* = 0.062, η^2^ = 0.190, power = 0.470); however, the IG demonstrated significantly faster post-intervention left cut completion times (*p* = 0.002, *g* = −1.57) compared to the CG. Moreover, large and significant improvements in left cut completion times were observed for the IG (*p* < 0.001, *g* = −1.63, −9.1%, −0.215 s) and CG (*p* = 0.002, *g* = −1.27, −5.0%, −0.123 s) (Table 3 and Table 4, Figure 4) post-intervention, but these improvements were greater than SDD for the IG only. Mean (*p* = 0.062, *g* = −0.89) and percentage (*p* = 0.038, *g* = −1.00) improvements were greater for the IG compared to CG, with moderate effect sizes (Figure 4). 

#### 3.1.3. Right COD Deficit (CODD)

Large and significant main effects for time were found for right CODDs (*p* < 0.001, η^2^ = 0.839, power = 1.000). In addition, a large and significant interaction effect of time and group for right CODD was also observed (*p* = 0.025, η^2^ = 0.262, power = 0.639), with the IG displaying significantly shorter post-intervention right CODDs (*p* = 0.001, *g* = −1.79) compared to the CG. Moreover, a very large and large, significant improvement in right CODD was observed for IG (*p* = 0.001, *g* = −2.43, −51.5%, −0.252 s) and CG (*p* < 0.001, *g* = −1.32, −27.9%, −0.145 s) post-intervention, respectively (Table 3 and Table 4, Figure 5). These changes were greater than the SDD for the IG only. Mean (*p* = 0.025, *g* = −1.09) and percentage (*p* = 0.003, *g* = −1.57) improvements were significantly greater for the IG compared to the CG, with moderate and large effect sizes (Figure 5), respectively.

#### 3.1.4. Left CODD

Large and significant main effects for time were found for left CODDs (*p* < 0.001, η^2^ = 0.606, power = 0.998). Although a small, non-significant interaction effect of time and group for left CODD was observed (*p* = 0.316, η^2^ = 0.059, power = 0.164), the IG displayed significantly shorter post-intervention left CODDs (*p* = 0.011, *g* = −1.27) compared to the CG. Moreover, a large and moderate, significant improvement in left CODDs was observed for the IG (*p* = 0.012, *g* = −1.63, −39.1%, −0.187 s) and CG (*p* = 0.013, *g* = −1.17, −21.7%, −0.124 s) post-intervention, respectively (Table 3 and Table 4, Figure 6). These changes were greater than the SDD for the IG only. Although non-significant, mean (*p* = 0.316, *g* = −0.46) and percentage (*p* = 0.062, *g* = −0.78) improvements were slightly greater for the IG compared to the CG, with a small and moderate effect size (Figure 6), respectively.

#### 3.1.5. The 10 m Sprint

No significant main effects (*p* = 0.400, η^2^ = 0.042, power = 0.129) or interactions (*p* = 0.367, η^2^ = 0.048, power = 0.141) were observed for 10 m sprint performance, and no significant differences in post-intervention 10 m sprints times were demonstrated between groups (*p* = 0.390, *g* = −0.39). The IG displayed a non-significant, yet small improvement in 10 m sprints following the intervention (*p* = 0.328, *g* = −0.29, −1.4%, −0.027 s), while a non-significant, trivial difference was observed for the CG (*p* = 0.957, *g* = 0.01, +0.1%, +0.001 s) (Table 3 and Table 4). These changes, however, were not greater than the SDD for both groups. Although non-significant, mean (*p* = 0.374, *g* = −0.41) and percentage (*p* = 0.370, *g* = −0.41) improvements were slightly greater for the IG compared to CG, with a small effect size.

#### 3.1.6. CMAS Right

Large and significant main effects for time were found for right CMAS (*p* = 0.041, η^2^ = 0.224, power = 0.551). In addition, a large, significant interaction effect of time and group for right CMAS was also observed (*p* = 0.018, η^2^ = 0.287, power = 0.694), with the IG showing slightly lower post-intervention right CMASs (*p* = 0.001, *g* = −0.26) compared to the CG. A moderate and significant improvement in right CMAS was observed for the IG (*p* = 0.025, *g* = −0.85, −22.5%, −1.46 score) following the intervention, whereas the CG demonstrated no significant and trivial changes in right CMAS post-intervention (*p* = 0.779, *g* = 0.08, +5.6%, +0.12 score) (Table 3 and Table 4, Figure 7). These changes were greater than the SDD for the IG only. Although non-significant, mean (*p* = 0.122, *g* = −1.16) and percentage changes (*p* = 0.089, *g* = −1.12) were greater for the IG compared to CG, with moderate effect sizes (Figure 7).

#### 3.1.7. CMAS Left

Large and significant main effects for time were found for left CMAS (*p* = 0.015, η^2^ = 0.302, power = 0.725). In addition, a large and significant interaction effect of time and group for left CMAS was also observed (*p* = 0.001, η^2^ = 0.499, power = 0.972), with the IG showing lower post-intervention left CMASs (*p* = 0.318, *g* = −0.77) compared to the CG. A large, significant improvement in left CMAS was observed for the IG (*p* = 0.018, *g* = −1.46, −33.9%, −2.21 score) following the intervention, whereas the CG demonstrated no significant and small changes in left CMAS post-intervention (*p* = 0.306, *g* = 0.33, +18.5%, +0.45 score) (Table 3 and Table 4). These changes were greater than the SDD for the IG only. Although non-significant, mean (*p* = 0.089, *g* = −1.83) and percentage changes (*p* = 0.137, *g* = −1.51) were substantially greater for the IG compared to CG, with large effect sizes (Figure 8).

### 3.2. Task Specific Changes in Cutting Movement Quality: CMAS Deficit Changes

Task-specific pre-to-post changes in cutting movement quality for the IG and CG are presented in Appendix A. In general, the IG demonstrated lower incidences/frequencies of CMAS deficits post-intervention, with improvements in cutting movement quality and reductions in potentially hazardous deficits including lateral trunk flexion, extended knee postures, knee valgus, hip internal rotation, and improved PFC braking strategies (Appendix A). Conversely, changes in the frequencies of CMAS deficits were smaller for the CG, and in some cases, frequencies of CMAS deficits increased, including lateral trunk flexion, extended knee postures, trunk leaning back, and foot position (Appendix A).

## 4. Discussion

The aim of the present study was to determine the effects of a six-week COD speed and technique modification training intervention on cutting performance and movement quality in male youth soccer players. The primary findings were that six-weeks COD speed and technique modification training performed during the competition phase, in addition to normal skills and strength training, produced meaningful improvements in cutting performance times, CODDs, and cutting movement quality (i.e., lower CMAS) in male youth soccer players (Table 3 and Table 4, Figure 3, Figure 4, Figure 5, Figure 6, Figure 7 and Figure 8), thus supporting the study hypotheses. The observed improvements in performance times and CODDs for the IG were, on average, two-times greater than the CG, who also continued their normal field-based warm-ups (Table 3 and Table 4). However, the CG demonstrated no meaningful or significant improvements in CMASs, in contrast to the IG who demonstrated meaningful improvements in cutting movement quality.

As cutting actions are frequently performed manoeuvres in soccer [1,2] and linked to decisive moments (i.e., assists and goal scoring) [3], the ability to cut rapidly can be considered an important quality to develop. The results of the present study substantiate previous research that found COD speed training interventions improved COD speed completion times in male youth soccer players [6,7,8], with the present study finding COD speed and technique modification training resulted in meaningful improvements in cutting completion time (*p* < 0.001, *g* = 1.63–1.90, ~9–11%), and these changes were greater than the CG (Table 3 and Table 4, Figure 3 and Figure 4). Moreover, CODD has been recently developed and suggested to provide a more isolated measure of COD ability [34,35,36]. Previous studies, which have shown improvements in COD speed tasks, have only assessed completion times [6,7,8], and these tasks are mainly comprised of linear running and thus, biased towards athletes with superior acceleration and linear speed capabilities [34,35]. Conversely, to the best of our knowledge, the present study is the first to monitor changes in CODD in response to a training intervention in youth soccer players. Critically, substantial and meaningful improvements in CODD were demonstrated by the IG (*p* ≤ 0.012, *g* = −1.63 to −2.43, ~40–52%), which were approximately two times greater than the CG (Table 3 and Table 4, Figure 5 and Figure 6). These findings highlight the effectiveness of field-based COD speed and technique modification training in youth soccer athletes, which can be achieved in-season with two, twenty-minute sessions per week. This form of training can be simply and easily integrated into the warm-ups of field-based tactical/technical sessions in soccer, highlighting the applicability and feasibility of COD speed and technique modification training.

Cutting is a key action associated with non-contact ACL injuries in soccer [10,11,12] due to the propensity to generate large multi-planar joint loads that can strain and potentially rupture the ACL [16]. COD technique modification training has been shown to be an effective modality for addressing “high-risk” postures (lateral trunk flexion, lateral foot plant distance, knee flexion angle, internal foot rotation angles) and reducing potentially hazardous knee joint loading when assessed using 3D motion analysis [13,25,26]. Although the present study used a qualitative screening tool (CMAS) to monitor changes in cutting movement quality, the CMAS has been recently validated against the gold standard of 3D motion analysis [29,30] which presents as a more practical, less expensive screening tool to implement in applied sporting environments. Nevertheless, the results of this study confirm that COD speed and technique modification training (with feedback and externally directed verbal cues from a coach) resulted in meaningfully lower CMASs post-intervention, which were greater than the SDD (Table 3 and Table 4, Figure 7 and Figure 8) (*p* ≤ 0.025, *g* = −0.85 to 1.46, −23% to −34% vs. +6%–19%), while the CG remained unchanged. 

Similar to Stroube et al. [53], which assessed task-specific changes in the tuck jump assessment (i.e., changes in the frequencies of deficits demonstrated) following a neuromuscular training intervention with task-specific feedback, the IG in the present study demonstrated improved cutting movement quality and reductions in frequencies in “high-risk” deficits including lateral trunk flexion, extended knee postures, knee valgus, and improved PFC braking strategies (Appendix A). These findings are noteworthy because the aforementioned “high-risk” postures are associated with increased knee joint loading [14,22] which increases ACL [15,54] and PFP injury risk [55]. Furthermore, these postures are also characteristics of non-contact ACL injury [10,11,56]. As such, six-weeks COD speed training and technique modification with externally focused coaching cues and feedback from a coach is an effective training modality for reducing “high-risk” biomechanical and neuromuscular control deficits, and overall, improving movement quality in male youth soccer players. It is worth noting, however, that CMAS deficits are still demonstrated by youth soccer players (Appendix A) and thus, they should continue performing mitigation training interventions to address these deficits [20,28].

It has been suggested a “performance-injury trade-off” could exist when modifying the COD technique [14,24,27], whereby addressing “high-risk” postures could be detrimental to performance. While Dempsey et al. [13] reported reductions in knee joint loads due to changes in foot plant distance and trunk position, the authors failed to consider the implications of such changes on cutting performance (i.e., completion time, ground contact time, and exit velocity). Jones et al. [25] found changes in pivoting technique resulted in lower KAMs and faster completion times in netball players. However, these studies have used 3D motion analysis to monitor changes in COD technique and more importantly, these studies have not contained a CG; therefore, the results should be interpreted with caution. To the best of our knowledge, the present study is the first to consider the effect of COD speed and technique modification training on both performance (completion times, COD deficit) and injury-risk (CMAS) in comparison to a CG, using a field-based screening tool which was performed in a real-world setting. Notably, COD speed and technique modification training with feedback and external verbal coaching cues was effective in improving cutting performance and improving cutting movement quality (Table 3 and Table 4), and these were significantly and meaningfully greater than the CG and SDD (Figure 3, Figure 4, Figure 5, Figure 6, Figure 7 and Figure 8). Collectively, these findings highlight that COD speed and technique modification in youth soccer players is an effective training modality for enhancing performance and addressing movement deficits associated with increased knee joint loading and potential injury-risk. This finding is noteworthy because, since the introduction of the EPPP, injury rates in youth-soccer have increased three-fold [31] and as such, reducing biomechanical and neuromuscular risk factors in youth-soccer is considered highly important, particularly as these players are striving for professional contracts.

The COD speed and training intervention focused on modifying biomechanical deficits associated with increased injury-risk [14,22,27] and promoting techniques required for faster performance [14,22,23,24]. For example, the programme focused on several aspects: a wide foot-plant is required for medio-lateral propulsive impulse generation and subsequent exit velocity during cutting [22,24]; faster performance and lower knee joint loading has been associated with increased PFC braking forces [41]; and trunk lean towards the direction of travel and reduced lateral trunk flexion is associated with faster performance and reduced knee joint loads [14,22,23]. Moreover, knee valgus is also a hazardous “high-risk” posture [56] with no associated performance benefits [14,23,24] and thus, was a further desired technical change in response to the intervention. Consequently, the IG training programme consisted of COD drills and externally focused verbal coaching cues (“slam on the brakes early”, “push the ground away” and “lean/face towards the direction of travel”) to promote safer mechanics [42], promote faster performance [43], and facilitate better retention [42]. These cues were used in order to evoke technical changes to encourage PFC braking and trunk lean towards the direction of travel, which are techniques associated with faster performance and reduced knee joint loads, while medio-lateral propulsive impulse was also emphasised to promote faster exit velocities. Interestingly, post-intervention, the IG demonstrated lower CMASs (Table 3, Figure 7 and Figure 8), which can be attributed to reduced incidences of CMAS deficits, such as lateral trunk flexion, initial and excessive valgus, hip internal rotation, and lack of PFC braking (Appendix A), while a wide lateral foot plant remained unchanged. Although these were qualitative evaluations only, it is speculated that the IG demonstrated safer cutting mechanics, and due to the strong relationship observed between CMAS and peak KAM [29], theoretically, the IG may demonstrate lower knee joint loading, which subsequently reduces non-contact ACL [15,54,56] and PFP injury risk [17,55]. Further research is required to determine the effect of COD technique modification on knee joint loading and performance using 3D motion analysis to further substantiate this claim. 

### Limitations

It should be noted that the present study only investigated a 70° side-step cutting task from a short approach distance (5 m), and thus, is only reflective of low-entry velocity side-step cutting ability. As the biomechanical demands of COD are angle- and velocity-dependent [27,36], and other COD actions are also performed, such as crossover cuts, split-steps and pivots, further research is needed to determine the effects of COD speed and technique modification training on CODs from different angles and approach distances, while also investigating different types of COD actions. Additionally, it is worth noting that male youth soccer players were only investigated, as such caution is advised regarding the generalisation of these results to athletes from different athletic populations. Further insight is required into the effects of COD speed and technique modification training in different athletic populations where cutting is a highly prevalent action for performance and also, non-contact ACL injury, such as rugby, American football, and handball. Moreover, it should be noted that there were five dropouts for the IG (Figure 1) due to match-related injuries or illness. However, the present study still achieved a priori sample size statistical power recommendation and dropouts reflect the environment of performing an in-season training intervention in a “real-world” professional soccer setting. Finally, while improved cutting movement quality was demonstrated following the COD speed and technique modification training, it is unknown whether if and how long the improved motor skill performance can be maintained/ retained for, and is, therefore, a recommended area of future research.

## 5. Conclusions

Six-weeks COD speed and technique modification training performed in-season (with the use of external verbal coaching cues and feedback), in addition to normal skills and strength training, resulted in significant and meaningful improvements in cutting completion time and CODDs in youth soccer players. These changes in performance were approximately two-times greater than the CG and exceeded the SDD. Furthermore, COD speed and technique modification training resulted in lower CMASs and improved cutting movement quality, while the CG demonstrated no meaningful or significant changes in CMAS. The improvements in cutting movement quality observed for the IG were attributed to technical improvements (reductions in deficits), such as lateral trunk flexion, knee valgus, PFC braking, and internal hip rotation and overall cutting movement quality. These findings indicate that improvements in cutting performance and movement quality can be achieved in-season, in a real-world sporting environment. As such, practitioners working with male youth soccer players should consider implementing two, twenty-minute COD speed and technique modification training sessions a week, in addition to normal skills and strength training, for improvements in cutting movement quality and performance. 

## Figures and Tables

**Figure 1 sports-07-00205-f001:**
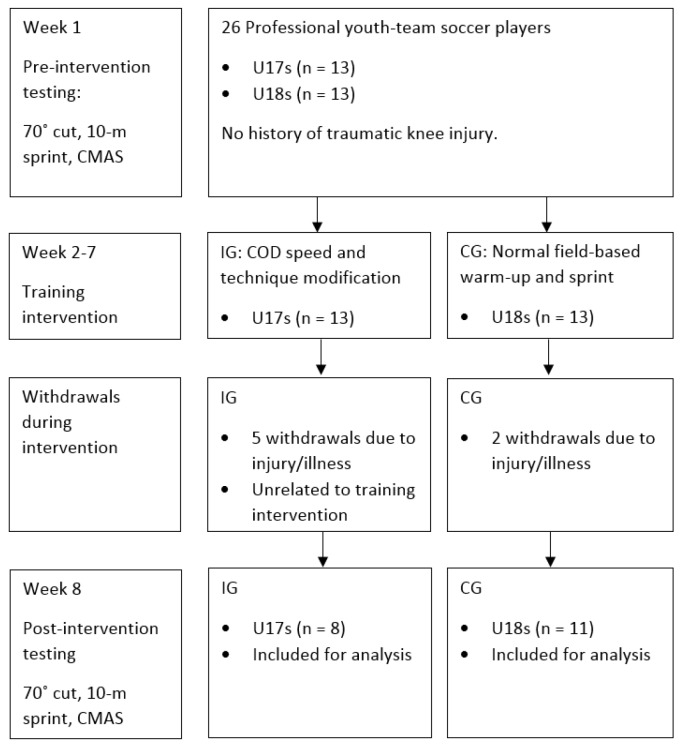
Flow diagram of participant participation throughout all stages of the intervention study. IG: Intervention group; CG: Control group; CMAS: Cutting movement assessment score.

**Figure 2 sports-07-00205-f002:**
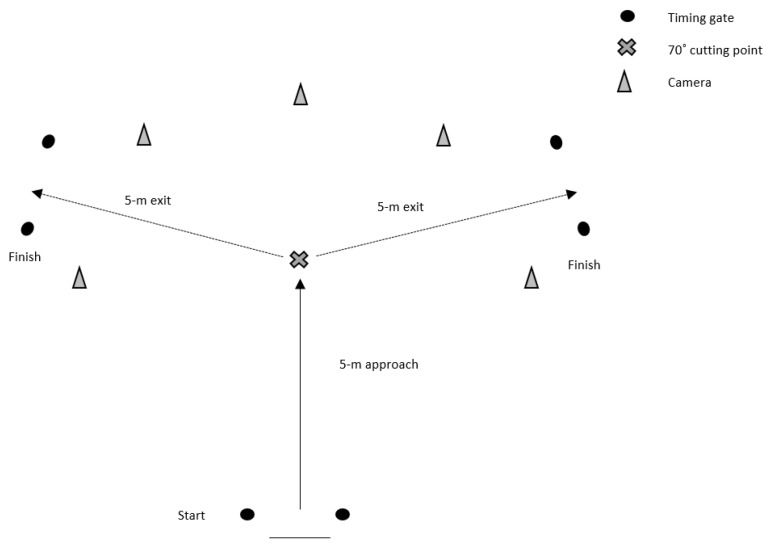
Schematic representation of the 70˚ cutting task.

**Figure 3 sports-07-00205-f003:**
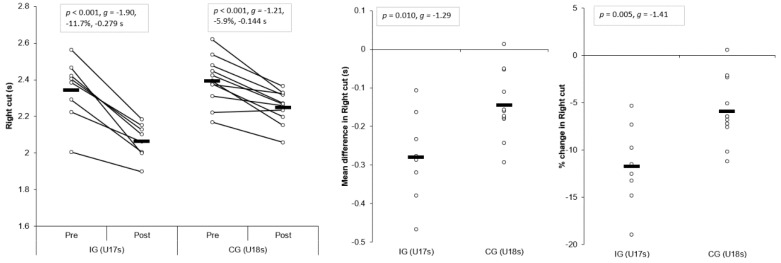
Individual plots illustrating pre-to-post changes in right cut completion times with individual mean and percentage changes; IG: Intervention group CG: Control group. Black rectangle denotes mean.

**Figure 4 sports-07-00205-f004:**
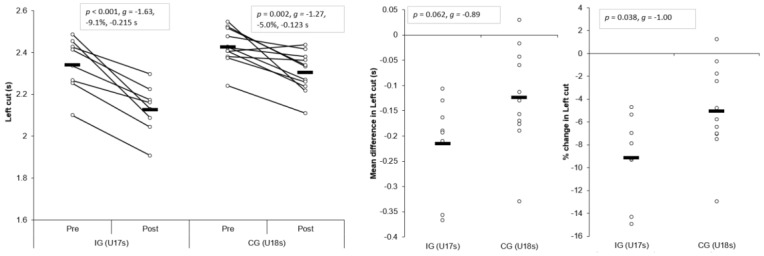
Individual plots illustrating pre-to-post changes in left cut completion times with individual mean and percentage changes; IG: Intervention group CG: Control group. Black rectangle denotes mean.

**Figure 5 sports-07-00205-f005:**
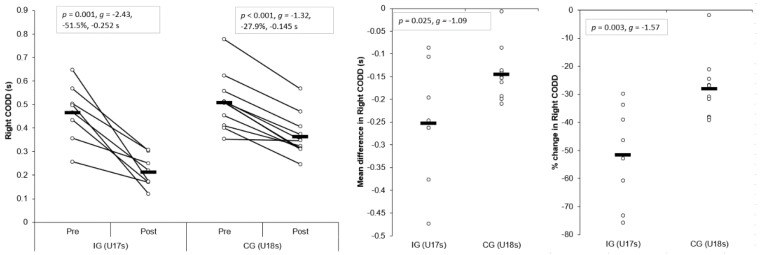
Individual plots illustrating pre-to-post changes in right CODDs with individual mean and percentage changes; IG: Intervention group CG: Control group; CODD: Change of direction deficit. Black rectangle denotes mean.

**Figure 6 sports-07-00205-f006:**
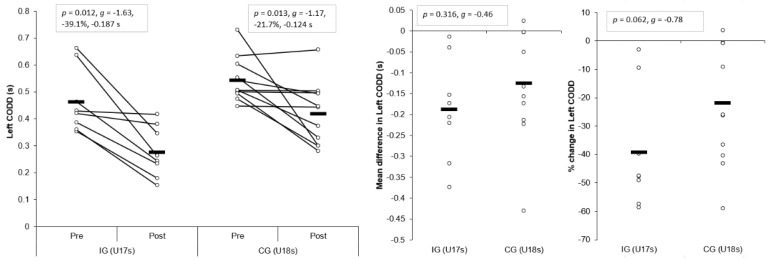
Individual plots illustrating pre-to-post changes in left CODDs with individual mean and percentage changes; IG: Intervention group CG: Control group; CODD: Change of direction deficit. Black rectangle denotes mean.

**Figure 7 sports-07-00205-f007:**
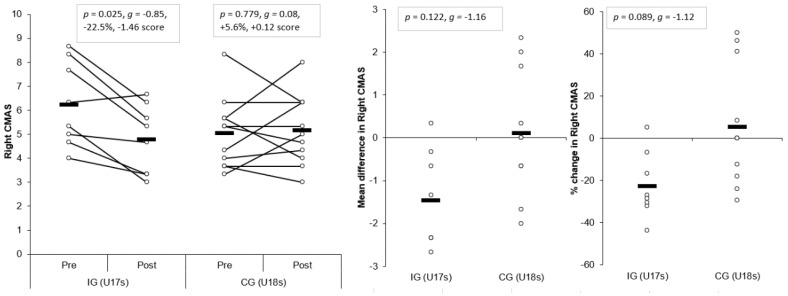
Individual plots illustrating pre-to-post changes in right CMAS with the individual mean and percentage changes; IG: Intervention group CG: Control group; CMAS: Cutting movement assessment score. Black rectangle denotes mean.

**Figure 8 sports-07-00205-f008:**
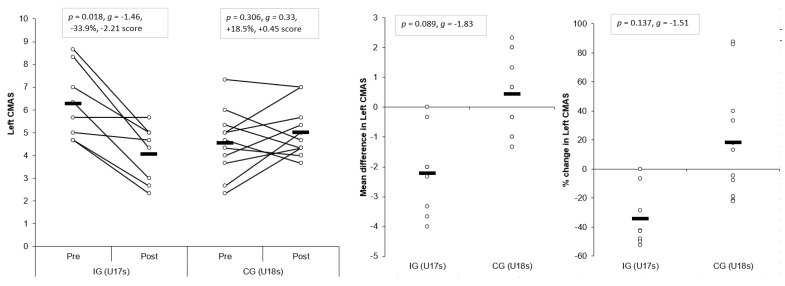
Individual plots illustrating pre-to-post changes in left CMAS with the individual mean and percentage changes; IG: Intervention group CG: Control group; CMAS: Cutting movement assessment score. Black rectangle denotes mean.

**Table 1 sports-07-00205-t001:** Within-session reliability measures.

IG Pre-Intervention within-Session Reliability										
**Variable**		**ICC**	**LB**	**UB**	**CV%**	**LB**	**UB**	**SEM**	**SDD**	**SDD%**
Completion time	Right Cut (s)	0.944	0.800	0.988	2.8	1.8	3.7	0.042	0.117	5.0
	Left Cut (s)	0.865	0.537	0.971	2.8	1.4	4.2	0.054	0.150	6.4
CODD	Right CODD (s)	0.889	0.622	0.976	15.1	8.3	21.9	0.043	0.120	25.9
	Left CODD (s)	0.843	0.461	0.966	13.9	8.6	19.3	0.055	0.153	33.0
Sprint	10-m sprint (s)	0.861	0.561	0.969	2.1	0.9	3.4	0.034	0.093	5.0
CMAS	CMAS Right	0.934	0.776	0.986	11.4	5.2	17.6	0.49	1.36	21.7
	CMAS Left	0.865	0.526	0.971	15.7	9.3	22.2	0.67	1.84	29.3
IG post-intervention within-session reliability										
Completion time	Right Cut (s)	0.934	0.786	0.985	2.0	1.6	2.4	0.026	0.071	3.5
	Left Cut (s)	0.917	0.732	0.982	2.3	1.1	3.4	0.037	0.102	4.8
CODD	Right CODD (s)	0.873	0.596	0.972	20.8	15.0	26.6	0.027	0.074	34.7
	Left CODD (s)	0.870	0.580	0.972	17.0	10.5	23.4	0.038	0.106	38.4
Sprint	10-m sprint (s)	0.932	0.776	0.985	1.8	0.7	2.9	0.027	0.076	4.1
CMAS	CMAS Right	0.870	0.553	0.972	18.3	12.4	24.3	0.59	1.63	33.9
	CMAS Left	0.817	0.373	0.961	22.2	16.0	28.3	0.63	1.74	42.6
CG pre-intervention within-session reliability										
Completion time	Right Cut (s)	0.878	0.671	0.964	3.0	2.1	3.9	0.051	0.141	5.9
	Left Cut (s)	0.693	0.180	0.909	2.9	1.9	4.0	0.061	0.168	6.9
CODD	Right CODD (s)	0.847	0.585	0.955	14.9	9.6	20.3	0.052	0.145	28.3
	Left CODD (s)	0.661	0.096	0.899	13.2	8.7	17.7	0.062	0.172	31.5
Sprint	10-m sprint (s)	0.762	0.311	0.931	2.3	1.7	3.0	0.035	0.096	5.1
CMAS	CMAS Right	0.898	0.723	0.970	14.4	9.9	18.8	0.52	1.43	28.3
	CMAS Left	0.877	0.668	0.964	18.5	9.8	27.3	0.56	1.55	33.9
CG post-intervention within-session reliability										
Completion time	Right Cut (s)	0.727	0.214	0.921	3.1	2.2	4.1	0.059	0.163	7.2
	Left Cut (s)	0.903	0.738	0.972	2.0	1.3	2.7	0.033	0.092	4.0
CODD	Right CODD (s)	0.721	0.197	0.920	18.8	14.1	23.6	0.059	0.163	44.7
	Left CODD (s)	0.932	0.816	0.980	11.6	7.4	15.7	0.032	0.089	21.3
Sprint	10-m sprint (s)	0.830	0.546	0.950	2.4	1.4	3.3	0.036	0.099	5.3
CMAS	CMAS Right	0.854	0.609	0.957	19.2	14.1	24.2	0.63	1.76	33.9
	CMAS Left	0.774	0.355	0.935	15.3	8.9	21.7	0.65	1.81	36.0

Key: IG: Intervention group; CG: Control group; CODD: Change of direction deficit; CMAS: Cutting movement assessment score; ICC: Intraclass correlation coefficients; CV %: Coefficient of variation; LB: Lower bound 95% confidence interval; UB: Upper bound 95% confidence interval; SEM: Standard error of measurement; SDD: Smallest detectable difference.

**Table 2 sports-07-00205-t002:** Intra- and inter-rater reliability for CMAS criteria and total score.

Variable/CMAS Tool Criteria	Intra-Rater Reliability		Inter-Rater Reliability	
	% Agreement	*k*	% Agreement	*k*
Clear PFC braking	92.3	0.755	94.7	0.894
Wide lateral leg plant	96.2	0.920	100.0	1.000
Hip in an initial internally rotated position	100.0	1.000	100.0	1.000
Initial knee ‘valgus’ position	96.2	0.866	89.5	0.789
Inwardly rotated foot position	96.2	0.922	100.0	1.000
Frontal plane trunk position relative to intended direction	96.2	0.935	94.7	0.906
Trunk upright or leaning back throughout contact	96.2	0.906	100.0	1.000
Limited Knee Flexion during final contact	96.2	0.923	94.7	0.872
Excessive Knee ‘valgus’ motion during contact	96.2	0.920	94.7	0.906
Average	96.2	0.905	96.5	0.930

Key: PFC: Penultimate foot contact; CMAS: Cutting movement assessment score.

**Table 3 sports-07-00205-t003:** IG pre-to-post changes.

Variable	IG Pre		IG Post			Hedges’ *g* Effect Size			% Change				Mean Difference			
	Mean	SD	Mean	SD	*p*	*g*	LB	UB	Mean	SD	LB	UB	Mean	SD	LB	UB
R Cut (s)	2.344	0.172	2.065	0.095	**<0.001**	−1.90	−3.08	−0.72	−11.7	4.3	−16.2	−7.7	−0.279	0.115	−0.375	−0.182
L Cut (s)	2.342	0.130	2.128	0.119	**<0.001**	−1.63	−2.76	−0.50	−9.1	3.8	−12.5	−5.8	−0.215	0.097	−0.296	−0.133
R CODD (s)	0.466	0.121	0.214	0.068	**0.001**	−2.43	−3.72	−1.14	−51.5	17.4	−72.8	−35.3	−0.252	0.129	−0.360	−0.144
L CODD (s)	0.464	0.121	0.277	0.095	***0.012***	−1.63	−2.76	−0.50	−39.1	21.2	−54.3	−26.3	−0.187	0.123	−0.290	−0.084
10-m sprint (s)	1.878	0.081	1.850	0.098	*0.328*	−0.29	−1.27	0.70	−1.4	3.9	−2.6	−0.4	−0.027	0.075	−0.034	0.090
R CMAS	6.3	1.8	4.8	1.4	***0.025***	−0.85	−1.88	0.17	−22.5	15.7	−31.5	−15.2	−1.5	1.1	−2.6	−0.4
L CMAS	6.3	1.6	4.1	1.2	***0.018***	−1.46	−2.57	−0.36	−33.9	20.3	−47.3	−22.9	−2.2	1.5	−3.5	−1.0

Key: R: Right; L: left; IG: Intervention group; CODD: Change of direction deficit; CMAS: Cutting movement assessment score; LB: Lower-bound 95% confidence interval; UB: Upper-bound 95% confidence interval. Note: italic denotes non-parametric.

**Table 4 sports-07-00205-t004:** CG pre-to-post changes.

Variable	CG Pre		CG Post			Hedges’ *g* Effect Size				% Change			Mean Difference			
	Mean	SD	Mean	SD	*p*	*g*	LB	UB	Mean	SD	LB	UB	Mean	SD	LB	UB
R Cut (s)	2.395	0.131	2.251	0.089	**<0.001**	−1.21	−2.28	−0.15	−5.9	3.5	−8.3	−3.7	−0.144	0.089	−0.203	−0.084
L Cut (s)	2.429	0.087	2.306	0.097	**0.002**	−1.27	−2.34	−0.19	−5.0	3.9	−7.1	−3.1	−0.123	0.099	−0.190	−0.057
R CODD (s)	0.510	0.117	0.365	0.088	**<0.001**	−1.32	−2.41	−0.24	−27.9	10.5	−38.3	−18.5	−0.145	0.057	−0.183	−0.106
L CODD (s)	0.545	0.082	0.420	0.116	0.013	−1.17	−2.23	−0.11	−21.7	21.4	−30.8	−14.9	−0.124	0.136	−0.216	−0.032
10-m sprint (s)	1.885	0.058	1.885	0.075	0.957	0.01	−0.97	0.99	0.1	3.3	−0.9	1.0	0.001	0.061	−0.041	−0.040
R CMAS	5.1	1.5	5.2	1.5	*0.779*	0.08	−0.90	1.06	5.6	28.2	−11.1	22.3	0.1	1.4	−0.9	1.1
L CMAS	4.6	1.4	5.0	1.1	*0.306*	0.33	−0.65	1.32	18.5	39.8	−5.0	42.0	0.5	1.3	−0.5	1.4

Key: R: Right; L: left; CG: Control group; CODD: Change of direction deficit; CMAS: Cutting movement assessment score; LB: Lower-bound 95% confidence interval; UB: Upper-bound 95% confidence interval. Note: italic denotes non-parametric.

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
