# Peer review of "The Effects of Six-Weeks Change of Direction Speed and Technique Modification Training on Cutting Performance and Movement Quality in Male Youth Soccer Players"

_sports, 2019, doi:10.3390/sports7090205_

Round 1

Reviewer 1 Report

I want to thank the authors and the editor for giving me the chance of reviewing this work. Its aim is to determine the effects of a six-week change of direction (COD) speed and technique modification training intervention on cutting performance and movement quality in male youth (U17-U18s) (U17-U18s).

I find this work very interesting because the potential readers of this work will find: 1) a clear evidence on why implement COD speed and technique modification training in youth players (although further works are required to set cause-and-effect as only 8 players completed the intervention); and 2) practical exercises to improve COD technique and performance in trainings [supplement 1]. All in all, there are some issues that must be address before considering this work for publication in Sports

Please, find my comments in the attached document

Author Response

Reviewer 1

I want to thank the authors and the editor for giving me the chance of reviewing this work. Its aim is to determine the effects of a six-week change of direction (COD) speed and technique modification training intervention on cutting performance and movement quality in male youth (U17-U18s) (U17-U18s).

I find this work very interesting because the potential readers of this work will find: 1) a clear evidence on why implement COD speed and technique modification training in youth players (although further works are required to set cause-and-effect as only 8 players completed the intervention); and 2) practical exercises to improve COD technique and performance in trainings [supplement 1]. All in all, there are some issues that must be address before considering this work for publication in Sports

Response: Thank you for your kind words.

General comment 

I  acknowledge the work done by the authors in the literature review. However, I think 180 references is quite a lot. The usual number for this kind of papers is between 20 and 40 with most of them 30 or less. I advise the authors to try to reduce the number of references as many as possible.

Response: Thank you for your comment. Apologies, we have now reduced this to 56 references.

Introduction

Line 38. Because it is the first time in the main text in which the acronym COD is used. The authors should write “change of direction (COD)”.

Response: Thank you for your comment. This has been amended. Please see line 41.

I find the introduction very interesting because provide quite relevant learning to the reader. However, I think it is a bit long and I encourage the authors to reduce it (whenever possible). As such, they can do:

Remove the first sentence (lines 37-39). It does not provide significant value to the introduction Reducing some of the paragraphs by summarising the information Reducing the number of references and deleting the non-essential statements

Response: Thank you for your comment. Following your suggestions, we have now reduced the introduction and tried to be more concise. As such, the word count has changed from 1459 to 996 words and reduced the number of references.

Methodology

 Research design

How much time did the warm-up of the control group (U18 players) last?

Response: Thank you for your comment. We have added the following to line132 “(i.e. 5 minutes mobilisation exercises before progressing to 15 minutes low-level jump-landing and sprint drills)”.

Just to clarify, warm-up only consisted of low-level jump-landing and sprint drills? So, no joint mobility or drills with the ball were performed, ?

Response: Thank you for your comment. As stated above, they did perform 5 minutes mobilisation, but no ball work was performed in the warm-up.

I did not totally understand how the training-effect was controlled (the authors state the following in the next section) Line 172-173: All participants participated in the same skills sessions, five times a week (~90 minutes sessions). Lines 173-174: All participants performed two resistance training sessions a week, and received the same training programmes

Response: Thank you for your comment. The intervention only replaced two warm-ups of the 5 soccer sessions (where they would complete their normal warm-ups) where they performed the specific COD training. However, both groups performed their technical/tactical soccer sessions together (i.e. same volume) and also performed the same strength training (i.e. resistance training) where (sets, reps, and intensity were controlled and the same).

What are the skill sessions about? The regular training session of each team? Please explain how they were controlled (i.e. run by the authors; supervision of the authors; the coaches agreed on designing the same training for both groups). Also, I suggest the authors to explain more what the skill sessions were about (i.e. technical exercises, small- sided games, etc.

Response: Thank you for your comment. Apologies, this has been amended. It now reads as follows “All participants participated in the same technical and tactical soccer sessions (led by head soccer coach), five times a week (~90 minutes sessions on match day +2, +3, -3, -2, -1).” Line 147-148.

What is resistance training? was it an additional training for players? There was two resistance training a week. Were they different days than the skill session? Please explain this

Response: Thank you for your comment. Resistance training is strength training. We have now changed it to strength training throughout. These were two additional strength training sessions (performed in a gym) on match day +2 and match day -3, where they performed these sessions in the morning, and soccer practice in the afternoon of that day.

I understand that participants also played a weekly match during the study, so they were playing soccer/training 7 days a week? This is very important, because according to the authors, control group played in a different team and it is important o know up to what extent this kind of variables were controlled.

Response: Thank you for your comment. No, the soccer players performed 5 soccer training sessions per week, one match day, and had one rest day. The weekly schedule is presented below. And for clarification, the players from both teams were integrated and performed the same soccer technical/tactical sessions.

Match day

Match day +1

Match day +2

Match day +3

Match day -3

Match day -2

Match day -1

Morning

Match

Rest day

Strength

COD technique & Soccer

Strength

COD technique & Soccer

Soccer

Afternoon

Soccer

Soccer

Participants

I understand that the first squad of the club played in 5th tier in English football league by the time of the study, right? But players participating in this study played in a league specific for U17 players and U18 Am I right?

I am not totally up to date of the English competitive league system in youth soccer. I think that the top league in U18 players is U18 Premier League which is divided into two divisions, but the clubs participating in these leagues are in premier league or championship (at least most of them) so I understand that the participants of this study are not playing in the top league in the UK, right? Therefore, could the authors explain in the paper the playing level of the participants? I think that it is an interesting demographic data to report.

Response: Thank you for your comment. Yes, you are correct. Although the first team (senior team) played in the 5th tier of English football, the U17s and U18s played in a regional league against youth-teams from teams of a similar playing level). The team also played in the FA youth cup, reaching the 2nd round, and represented Manchester County winning the national cup. We have added this information to line 123-124.

I suggest the authors move the second paragraph of the “participants” subsection to the end of the Response: Thank you for your comment. This paragraph has been moved. Figure 1: the authors state that injuries/illness responsible for 5 withdrawals were unrelated to the training Could they provide further information to support this statement?

Response: Thank you for your comment. By this, we were trying to emphasise that no injuries were sustained during these intervention sessions. We have added “match-related” injuries to line 141 but removed unrelated to the training intervention in text. We hope this is clearer.

Procedures and field tests

Why was the field-testing performed in an indoor hardwood court instead of on the surface used for the players when playing soccer or in a soccer surface (artificial turf / natural grass)? Explain also in the

Response: Thank you for your comment. Unfortunately, due to logistical and facility constraints out of control of the researchers, the synthetic field-turf was unavailable on the dates we were given permission by the head coach and sports science staff to perform the testing. The team perform their training at two different locations and share the field-turf with a high-school, and therefore have limited access and availability. We appreciate that the surface is a limitation; however, given this was the only time we were given to perform the pre-and post-testing, we thought that it would still be adequate because it provides a controlled and standardised environment.

Illness/injury: Evidence that the illness is not related to the intervention (figure 1)

Response: Thank you for your comment. We have answered this above and removed “unrelated to the training intervention” .

Why field testing was performed on a hardwood court??

Response: Thank you for your comment. We have answered this above.

70º cutting task

Line 213: I suggest the authors be as much concise as Thus, I suggest them to remove the first part of the sentence “given the importance of cutting ability in soccer” it is already stated in the introduction. In methods, the authors must go straight to the point.

Response: Thank you for your comment. We have removed this and tried to be more concise throughout the manuscript.

Figure 2: I suggest the authors move Figure 2 and place it at the end of this section

Response: Thank you for your comment. We have moved Figure 2, as per your recommendations.

Results

 Line 364: there is a typo with the tables’ names. I think they are Table 4 and 5

Response: Thank you for your comment. Apologies for this error. This has been amended.

Line 365-366: remove the sentence “the following pre-to-post changes will be …”

Response: Thank you for your comment. This has been deleted.

Adverse events: I recommend the authors to delete this subsection. It has already been stated in Figure 1.

Response: Thank you for your comment. We have removed this.

Discussion:

Sometimes the authors provide too many references for the same statement (i.e. line 499 or line 502). Because 180 references are too many, I recommend the authors to avoid In the given example I think that one or two references are enough. There is no need to have four references. All in all, I acknowledge the author’s effort Response: Thank you for your comment. As stated earlier, we have reduced the number of references.

Lines 516-520: add the outcomes from the validation of the CMAS against the gold standard so the reader can know how reliable the CMAS is

Response: Thank you for your comment. We have added this information in the introduction, thus we feel it would be unnecessary to repeat this information again.

Conclusions:

Line 609: delete “in conclusion”

Response: Thank you for your comment. This has been amended.

Reviewer 2 Report

This paper attempted to elucidate if 6 weeks of directed change of direction training elicited performance alteration in soccer player. Overall the authors should be applauded for a well-designed and well written study. It was refreshing to read a highly detailed methods section for this manuscript. I have minor comments to help with readability and structure but I recommend this paper accepted with minor revisions. I will detail line by line comments below.

Major points of concern

The introduction reads more of a literature review than and introduction for a research article. My suggestion would be to merge paragraphs 82-122 and be as succinct as possible. This information is good but could be reduced to two high quality paragraphs.  

Minor concerns

Line 40: please specify the time frame of the 609 cuts. Is this per season, training or match play? Currently, this is unclear.

Line 223: 6 cutting trails were performed at maximum? Please specify  

Line 351: Table 2 should be table 1. Please adjust other contents within paper to reflect these changes

Line 364: Tables numbers should be changed to Table 4 and Table 5

Line 379: due to the large amount of data in the tables it is suggested to provide highlights on important p values to draw the reader’s attention.

Line 546: the sentence is incomplete at the end. Please remove the word “while”

Line 549: change author to authors.

Author Response

Reviewer 2

This paper attempted to elucidate if 6 weeks of directed change of direction training elicited performance alteration in soccer player. Overall the authors should be applauded for a well-designed and well written study. It was refreshing to read a highly detailed methods section for this manuscript. I have minor comments to help with readability and structure but I recommend this paper accepted with minor revisions. I will detail line by line comments below.

 Response: Thank you for your kind words.

Major points of concern

The introduction reads more of a literature review than and introduction for a research article. My suggestion would be to merge paragraphs 82-122 and be as succinct as possible. This information is good but could be reduced to two high quality paragraphs.  

Response: Thank you for your comment. Your point was raised by reviewer 1; thus, we have reduced the length of the introduction from 1459 to 996 words, reduced from 7 to 6 paragraphs,  and reduced the number of references.

Minor concerns

Line 40: please specify the time frame of the 609 cuts. Is this per season, training or match play? Currently, this is unclear.

Response: Thank you for your comment. Apologies for this oversight. We have amended this. Please see line 37-39.

Line 223: 6 cutting trails were performed at maximum? Please specify  

Response: Thank you for your comment. Apologies for this oversight. We have amended this. Please see line 187.

Line 351: Table 2 should be table 1. Please adjust other contents within paper to reflect these changes

Response: Thank you for your comment. Apologies for this oversight. We have amended this throughout.

Line 364: Tables numbers should be changed to Table 4 and Table 5

Response: Thank you for your comment. Apologies for this oversight. We have amended this.

Line 379: due to the large amount of data in the tables it is suggested to provide highlights on important p values to draw the reader’s attention.

Response: Thank you for your comment. Apologies, but we do not fully understand this point. Do you mean to highlight in the table the important p values? If so, we have now bolded the statistically significant p values.

Line 546: the sentence is incomplete at the end. Please remove the word “while”

Response: Thank you for your comment. This has been deleted.

Line 549: change author to authors.

Response: Thank you for your comment. This has been amended. Please see line 493.

Round 2

Reviewer 1 Report

I want to congratulate the authors for the improvement they have done in their manuscript. They not only have addressed all the suggestions from both reviewers but have significantly improved the quality of their manuscript.

After a careful reading of this second version, I think the manuscript has now reached a sufficient quality to be published in Sports. However, there are some minor concerns I would like to discuss with the authors.

Line 127: The authors used the G*Power to estimate the minimum sample size required to achieve a power of 0.80 based on "a prior analysis". Could the authors provide further information about the statistical test selected for the analysis? T-Test, F-Test, etc. and explain the reason.
Lines 123-125: The description of the sample can be improved: Players belong to an “Under 18 team playing in a regional league by the time of the study and participating in FA youth cup” Besides, the first team of the club played in “5th tier in English football league

* It is important to know the playing level of the first team of the club. But it is more important to know the level of the players, and it seems to be quite high. Maybe not the top in the UK, but they are not amateur youth players at all.
 Line 148: Although I understand what +1, +2, +3, -1, -2, -3 mean, I think some potential readers can mislead this information. I think if the authors add “day” after each number (i.e. +1 day, +2 days, +3 days, -1 day, -2 days, -3 days) all potential readers will understand it for sure. Thus, I suggest the authors add “day” after the numbers, at least the first time they appear in the manuscript.

Besides these new comments, I do not have further feedback to provide to the authors to improve the quality of their manuscript. Once again, I congratulate the authors for the improvements, and I appreciate the editors the opportunity of reviewing this work.

Author Response

I want to congratulate the authors for the improvement they have done in their manuscript. They not only have addressed all the suggestions from both reviewers but have significantly improved the quality of their manuscript.

Response: Thank you for your kind words.

After a careful reading of this second version, I think the manuscript has now reached a sufficient quality to be published in Sports. However, there are some minor concerns I would like to discuss with the authors.

Line 127: The authors used the G*Power to estimate the minimum sample size required to achieve a power of 0.80 based on "a prior analysis". Could the authors provide further information about the statistical test selected for the analysis? T-Test, F-Test, etc. and explain the reason. 

Response: Thank you for your comment. We have now added “(dependent T-Test)” on line 127, and have added the following for the rationale on line 130-132: “This approach was in line with recommendations for estimating sample sizes in strength and conditioning research based on effect sizes for pre-to-post designs [33].”

We followed the recommendations for estimating sample sizes in strength and conditioning research by Beck (2013). It is detailed how to use G*Power and effect sizes for pre-to-post designs for dependent t-tests. We hope this is satisfactory.

Beck, T. W. (2013). The importance of a priori sample size estimation in strength and conditioning research. The Journal of Strength & Conditioning Research27(8), 2323-2337.

Lines 123-125: The description of the sample can be improved: Players belong to an “Under 18 team playing in a regional league by the time of the study and participating in FA youth cup” Besides, the first team of the club played in “5th tier in English football league

* It is important to know the playing level of the first team of the club. But it is more important to know the level of the players, and it seems to be quite high. Maybe not the top in the UK, but they are not amateur youth players at all.

Response: Thank you for your comment and recommendations. We have added the extra details on line 123-126. It now reads as follows: “26 male youth soccer players from an English professional soccer club (Under 18s team, at the time of study, played in a regional league against youth teams of a similar standard, participated in the FA youth cup, and represented Manchester county; first team played in 5th tier in English football league at the time of the study) were recruited and participated in this study.”

Line 148: Although I understand what +1, +2, +3, -1, -2, -3 mean, I think some potential readers can mislead this information. I think if the authors add “day” after each number (i.e. +1 day, +2 days, +3 days, -1 day, -2 days, -3 days) all potential readers will understand it for sure. Thus, I suggest the authors add “day” after the numbers, at least the first time they appear in the manuscript.

Response: Thank you for your comment and recommendations. We have added the term “days”. Please see lines 152-154 and 240-241.

Besides these new comments, I do not have further feedback to provide to the authors to improve the quality of their manuscript. Once again, I congratulate the authors for the improvements, and I appreciate the editors the opportunity of reviewing this work.

Response: Thank you for your kind words and feedback. It has resulted in a much-improved manuscript.